# Identification of New Prognostic Genes and Construction of a Prognostic Model for Lung Adenocarcinoma

**DOI:** 10.3390/diagnostics13111914

**Published:** 2023-05-30

**Authors:** Xueping Chen, Liqun Yu, Honglei Zhang, Hua Jin

**Affiliations:** Key Laboratory of Molecular Medicine and Biotherapy, School of Life Science, Beijing Institute of Technology, Beijing 100081, China; 3120201394@bit.edu.cn (X.C.); 3120225698@bit.edu.cn (L.Y.); 3120215694@bit.edu.cn (H.Z.)

**Keywords:** lung adenocarcinoma, TCGA, prognostic model, *HAVCR1*, A549

## Abstract

Lung adenocarcinoma (LUAD) is a rapidly progressive malignancy, and its mortality rate is very high. In this study, we aimed at finding novel prognosis-related genes and constructing a credible prognostic model to improve the prediction for LUAD patients. Differential gene expression, mutant subtype, and univariate Cox regression analyses were conducted with the dataset from the Cancer Genome Atlas (TCGA) database to screen for prognostic features. These features were employed in the following multivariate Cox regression analysis and the produced prognostic model included the stage and expression of *SMCO2*, *SATB2*, *HAVCR1*, *GRIA1*, and *GALNT4*, as well as mutation subtypes of *TP53*. The exactness of the model was confirmed by an overall survival (OS) analysis and disease-free survival (DFS) analysis, which indicated that patients in the high-risk group had a poorer prognosis compared to those in the low-risk group. The area under the receiver operating characteristic curve (AUC) was 0.793 in the training group and 0.779 in the testing group. The AUC of tumor recurrence was 0.778 in the training group and 0.815 in the testing group. In addition, the number of deceased patients increased as the risk scores raised. Furthermore, the knockdown of prognostic gene *HAVCR1* suppressed the proliferation of A549 cells, which supports our prognostic model that the high expression of *HAVCR1* predicts poor prognosis. Our work created a reliable prognostic risk score model for LUAD and provided potential prognostic biomarkers.

## 1. Introduction

Lung adenocarcinoma (LUAD) is the major mortality cause of lung cancer and creates a heavy burden on the global healthcare system [1,2]. In past decades, standard treatments including surgical resection, radiotherapy, chemotherapy, and immunotherapy have advanced considerably [3]. However, the main drawbacks of treatment are the lack of effective diagnostic tools in the early stage and the resistant reaction to therapy [4,5]. Lung tumor classification mainly depends on morphology which is supported by immuno-histochemistry and is then further assisted by molecular techniques. The information on molecular abnormalities is critical for providing personalized treatment. Treatment protocols considering the molecular and biomarker testing results have greatly improved diagnostic and therapeutic accuracy, which has led to the rapid decline in lung cancer deaths from 2013 to 2018 in the United States [6]. Targeted therapy is a promising strategy to extend the lifespans of advanced-stage patients. The pathological functions of *ALK*, *EGFR*, *KRAS*, *HER3*, and *MET* in LUAD were clarified and target medicines were developed which improved the overall 5-year survival rates in advanced-stage LUAD patients [7]. However, the prognosis of LUAD is still poor, with a 5-year survival rate of less than 15% [8]. Therefore, further exploration of LUAD is required, including seeking valuable molecular markers and constructing an effective model to provide new insights for the diagnosis and clinical treatment of LUAD patients.

With the accumulation of high throughput and diverse biological sequencing data, more comprehensive information has been provided to unveil the dynamic molecular network and its biological consequences [9]. Genomic mutations, variations in DNA copy number and DNA methylation, as well as the expression of mRNA, microRNA, and long noncoding RNA (lncRNA), could affect dynamic molecular networks and be involved in different biological processes, such as cancer development [10]. These factors have been seen to interact with each other and some of them could have synergic effects on cancer [11].

Therefore, we focused on gene expression and gene mutation types to identify prognostic biomarkers. In this study, first, we downloaded RNA sequencing, nucleotide variation, and clinical data for LUAD from the Cancer Genome Atlas (TCGA) database. Then, a series of bioinformatic analyses were performed to construct and validate a risk score predictive model. Finally, we explored the effect of the newly identified prognostic gene *HAVCR1* in the LUAD cell line A549. Our research will contribute to the identification of new therapeutic targets and the improvement of clinical outcomes of LUAD patients.

## 2. Materials and Methods

### 2.1. The Source of Materials

The gene expression, clinical information, and nucleotide variation data of LUAD were obtained from the Cancer Genome Atlas (TCGA, https://portal.gdc.cancer.gov/, accessed on 29 December 2020) database. Human A549 lung cancer cells were stored in our laboratory liquid nitrogen container. The qPCR primers and siRNA were synthesized by the Genewiz company (South Plainfield, NJ, USA).

### 2.2. Screening Differentially Expressed Genes (DEGs)

We used the “limma” package in R to normalize the gene expression first. Then, the R function “Wilcoxon signed-rank test” was applied to identify the DEGs in two different thresholds: a moderate threshold of |log_2_ fold change (FC)| > 1 and false discovery rate (FDR) of *p*-value < 0.05 and a strict threshold of |log_2_ FC| > 3 and FDR of *p*-value < 0.01. 

### 2.3. Screening Genes Whose Mutant Subtypes Were Meaningful to Overall Survival

The package “maftools” in R was used to visualize the mutant landscape based on the nucleotide variation data of LUAD patients. The patients’ mutant subtypes from the top 30 genes with most mutant sites underwent label encoder treatment and the resulting information was combined with patients’ survival status and survival time to form a matrix. The influence of different mutant subtypes of each gene on overall survival was analyzed by the “survival” package in R. The “survdiff” function in the “survival” package was used to calculate a Log-Rank test that compares two or more survival curves. The Log-Rank test is a parameter-free test, similar to the chi-square test, and a *p*-value < 0.05 represents a significant difference. 

### 2.4. Construction of the Prognostic Model for LUAD Patients

After removing the normal samples and the tumor samples without clinical information, a total of 478 tumor samples were incorporated into univariate Cox regression analysis. For the DEGs selected in a moderate threshold, the *p*-value of univariate Cox regression was set as 0.01 and the genes with a |hazard ratio − 1| > 0.5 were filtered. For the DEGs selected in a strict threshold, the *p*-value of the univariate Cox regression was set as 0.05 and the genes with a |hazard ratio − 1| > 0.1 were filtered. In these candidate genes, the genes with a high value of |hazard ratio − 1| or the genes which had been reported to be related to LUAD were used for downstream multivariate Cox regression analysis. 

The above-selected DEGs, mutant genes, and clinical information, including patients’ stage, gender, and age, composed a matrix for the construction of the prognostic model. Different mutant types of mutant genes were transformed into integer scores according to their degree of impact on the overall survival. After setting the seed, all tumor samples were randomly divided into a training group (*n* = 336) and a testing group (*n* = 142) at a ratio of 7:3 by the “createDataPartition” function in R. The “both” direction multivariate stepwise Cox regression was performed using the data of the training group in a mode that automatically rejects the prognostic factors if the factors could not improve the performance of the risk score model. The coefficients of the remaining prognostic risk factors were used to construct a prognostic risk score model. The prognostic risk score of a patient could then be calculated using the following formula:Risk score = β1 × Factor1 + β2 × Factor2 + … + βi × Factor i

Here, β(i) represents the regression coefficient of a prognostic factor, and Factor(i) represents the gene expression level, a gene mutation score, or the pathologic stage of a LUAD patient.

### 2.5. Assessment and Validation of the Prognostic Model

The result of the multivariate stepwise Cox regression was subjected to the “predict” function in R to calculate the risk scores for LUAD patients which were divided into a high-risk group and a low-risk group according to their median risk score [12]. Next, we drew a Kaplan–Meier survival curve for the low and high-risk groups to assess the predictive ability of the model. The “survdiff” function was utilized to define the difference between the two groups. The clinical information on the “days_to_new_tumor_event_after_initial_treatment” was extracted and subjected to disease-free survival analysis (DFS) by the Kaplan–Meier survival curve. The time-dependent receiver operating characteristic (ROC) analysis, provided with the “Survival ROC” package of R, was applied to measure the prognostic accuracy of the model. The ROC analysis was also used to assess the tumor recurrence based on the clinical information, “new_tumor_event_after_initial_treatment”. All these analyses were performed on both the training and testing group so as to verify the produced model. 

### 2.6. Gene Ontology (GO) Term Analysis

GO term analysis was performed to explore the biological functions of the 43 candidate genes selected by univariate Cox regression. The gene symbols were converted to entrezIDs by the “org.Hs.eg.db” package of R first. Then, the “enrichGO” function of the “clusterProfiler” package was responsible for the analysis and the terms with a *p*-value < 0.05 were filtered. 

### 2.7. Cell Experiments

#### 2.7.1. Design and Synthesis of siRNA

The expression of *CYP24A1* and *HAVCR1* in A549 cells was evaluated in the Human Protein Atlas (HPA) platform (https://www.proteinatlas.org, accessed on 10 January 2022). We designed small interfering RNAs (siRNA) for *CYP24A1* and *HAVCR1* using the Dharmacon SMART selection algorithm (https://horizondiscovery.com/en/ordering-and-calculation-tools/sidesign-center, accessed on 24 June 2022) and synthesized them through the Genewiz company. The sequences of the siRNAs were 5′-CCU ACG CCA CCA AUU UCG UUU-3′ (SiCtrl-S), 5′-ACG AAA UUG GUG GCG UAG GUU-3′ (SiCtrl-AS), 5′-GCA GAA GAU UUG AGG AAU AUU-3′ (Si*CYP24A1*-S), 5′-UAU UCC UCA AAU CUU CUG CUU-3′ (Si*CYP24A1*-AS), 5′-GCC AAU ACC ACU AAA GGA AUU-3′ (Si*HAVCR1*-S), and 5′-UUC CUU UAG UGG UAU UGG CUU-3′ (Si*HAVCR1*-AS).

#### 2.7.2. Cell Culture and Transfection

The A549 cells were cultured in DMEM medium (hyclone) containing 10% fetal bovine serum (FBS, hyclone in Logan, UT, USA) in a humidified 5% CO_2_ incubator at 37 °C [13]. Then, the A549 cells were seeded in 6-well plates and the transfection of siRNA was performed using Lipofectamine 2000 (Invitrogen, Waltham, MA, USA) when the cells covered more than 50% of the plates. For transfection, we first replaced the cell culture medium with 1.5 mL serum-free DMEM and incubated the cells for 2 h. Then, 5 µL of 20 µM siRNAs and 6 µL Lipofectamine 2000 were diluted with 250 µL Opti-Medium, mixed into one tube, and incubated for 20 min at room temperature. Next, the transfection mixture was added uniformly to the cells and the cells were incubated in the incubator. Four hours later, 200 µL of FBS was added into the wells to keep the proliferation of cells. 

#### 2.7.3. Quantitative RT-PCR after Knockdown of *CYP24A1* or *HAVCR1* in A549 Cells

Forty-eight hours later, we extracted the mRNAs from the control group and experimental groups to detect the knockdown effect by reverse transcription (RT) and quantitative PCR (QuantStudio 3 Real-Time, ThermoFisher in Waltham, MA, USA). The sequences of the qPCR primers were 5′-GAG AAG GCT GGG GCT CAT TT-3′ (GAPDH-F), 5′-AGT GAT GGC ATG GAC TGT GG-3′ (GAPDH-R), 5′-CAG TAG CCA CTT CAC CAT CTT-3′ (*HAVCR1*-F), 5′-CTG GTG GGT TCT CTC CTT ATT G-3′ (*HAVCR1*-R), 5′-CTT ACG CCG AGT GTA CCA TTT A-3′ (*CYP24A1*-F), and 5′-GTG GCC TGG ATG TCG TAT TT-3′ (*CYP24A1*-R).

#### 2.7.4. CCK8 Proliferation Assay after Knockdown of *CYP24A1* or *HAVCR1* in A549 Cells

The A549 cells were transferred with SiCtrl, Si*CYP24A1*, or Si*HAVCR1* using Lipofectamine 2000 in 96-well plates and five replicate wells for each siRNA were generated. After 24, 48, 72, and 96 h of transfection, the cells were added to DMEM medium including 10% CCK-8 solution and incubated for 0.5 hr at 37 °C. Cell proliferation was detected by reading the OD values from a microplate reader (Cytation3, BioTek in Winooski, VT, USA) at a 450 nm wavelength [14].

## 3. Results

It has been well established that the main causes of cancers are accumulated genome mutations and the mis-expression of genes. Not all mutations and mis-expression events equally contribute to cancer development, that is, some of them affect the progress of cancers much more significantly than others. Therefore, the identification of these critical elements is important for cancer prognosis. To identify these factors and construct a valuable prognostic model based on these factors, we designed our study as follows (Figure 1). First, we downloaded RNA sequencing, nucleotide variation, and clinical data for LUAD from the TCGA database. Next, DEGs and survival-related mutant genes were selected and integrated with the clinical characteristics. Then, univariate Cox regression was used to identify the survival-related DEGs. Moreover, all samples were randomly divided into training groups and testing groups and the training group was applied to screen the prognostic factors, which were used to construct a prognostic risk score model later by using multivariate Cox regression. Furthermore, a comprehensive analysis was performed to assess the performance of the model. Finally, we explored the effect of the newly identified *HAVCR1* on cell proliferation in LUAD cell line A549.

### 3.1. Identification of Differentially-Expressed Genes 

The gene expression matrix of LUAD patients was downloaded from the TCGA database and contained 59 normal samples and 535 tumor samples. After normalizing the data, the DEGs between normal and tumor samples were identified from 34,478 genes. We aimed to construct a prognostic model containing statistically and functionally meaningful genes, therefore, two thresholds were set. A total of 10,107 DEGs, including 5004 down-regulated and 5103 up-regulated genes, reached the standard |log_2_ fold change (FC)| > 1 and false discovery rate (FDR) of *p*-value < 0.05 (Figure 2A, Appendix A), while 1352 DEGs met the criteria |log_2_ FC| > 3 and FDR of *p*-value < 0.01 and consisted of 372 down-regulated and 981 up-regulated genes (Figure 2B, Appendix A). The DEGs, which were identified to have more than two or eight times of expression change between normal and LUAD cancer tissues, were subjected to univariate Cox regression analysis.

### 3.2. Different Mutant Subtypes of TP53 and KEAP1 were Significantly Related to Prognosis

Varying types of genomic mutations might have different impacts on gene function and expression, and further cause various cellular effects. So, we next looked at the top 30 frequently mutated genes in LUAD using TCGA data. As displayed in the oncoplot, the top five genes with the highest mutation rate were *TP53*, *TTN*, *MUC16*, *CSMD3*, and *RYR2* and the mutation rates of the top 30 genes ranged from 16% to 48%. The oncoplot revealed that the foremost mutation type of the top 30 genes in all patients was the missense mutation and the second one was a nonsense mutation, indicating that amino acid substitutions and premature stop codons happen most often and probably affect protein function. In addition, it was found very often that one gene had multiple mutations. In missense mutation, C>A was the most frequent mutation event, followed by C>T change (Figure 3A). The prognostic value of different mutant subtypes of the top 30 frequently mutated genes was assessed by the survival package in R, and *TP53* and *KEAP1* were selected because their *p*-values were 0.002 and 0.004, respectively. The mutations of *TP53* and *KEAP1* have been proven to have prognostic value in LUAD. In addition to this, we found that different mutant subtypes unequally affected overall survival. Frame shift mutation resulted in the worst survival rate and nonsense mutation had relatively optimistic effects in both *TP53* and *KEAP1* (Figure 3B,C).

### 3.3. Construction of Prognostic Model for LUAD Patients

The univariate Cox regression analysis was conducted to investigate prognosis-related genes from the two groups of DEGs. For the group selected under a moderate condition, *OGFRP1*, *SATB2*, *GALNT4*, *FAM72A*, *SMCO2*, and *MIR99AHG* were picked up from 21 prognosis-related genes because of their high hazard ratio or known close relation with LUAD according to published research (Figure 4A, Appendix A) [15,16,17,18]. For the group selected under strict conditions, *ZIC5*, *GRIA1*, and *HAVCR1* were chosen from 22 prognosis-related genes according to their hazard ratio and biological functions (Figure 4B, Appendix A). To understand the function of the prognosis-related genes identified by univariate Cox regression analysis, we performed a GO enrichment examination using these genes. The results showed that these LUAD prognosis-related genes were mainly enriched in the regulation of G1/S transition of the mitotic cell cycle, cell polarity activities, G-protein coupled receptor pathway, PI3K pathway, and so on, consistent with the major roles of cell division factors in cancer development (Figure 4C). Combined with clinical information, a total of 13 factors, including age, gender, stage, expression of *ZIC5*, *GRIA1*, *HAVCR1*, *OGFRP1*, *SATB2*, *GALNT4*, *FAM72A*, *SMCO2*, *MIR99AHG*, and mutant subtypes of *TP53*, served as the candidate features for the LUAD prognostic model. Subsequently, the tumor samples were randomly divided into a training group and a testing group, and the training group was subjected to a multivariate stepwise Cox regression (Figure 4D and Table 1). According to the regression coefficients and the values of the selected features, the risk score of the prognostic model was established (Figure 4E). 

### 3.4. Assessment and Validation of the Prognostic Model

We evaluated our prognostic model by using an overall survival (OS) analysis. A total of 336 LUAD patients in the training group were separated, 168 into the high-risk group and 168 into the low-risk group, on the basis of median risk score. Likewise, 142 patients in the testing group were divided, with 70 in the high-risk group and 72 in the low-risk group. The survival curve from the training and testing groups indicated that the survival probability of high-risk groups was lower than that of low-risk groups (Figure 5A and Figure 6A). The disease-free survival curve showed similar trends (Figure 5B and Figure 6B). Receiver operating characteristic (ROC) analysis was applied to measure the exactness of the prognostic ability of the model. The results revealed that the area under the receiver operating characteristic curve (AUC) was 0.793 in the training group and 0.779 in the testing group, showing a favorable predictive performance (Figure 5C and Figure 6C). A total of 428 patients were with tumor recurrent information and 172 patients recurred during the follow-up time. In addition, the evidence that AUC was 0.778 in the training group and 0.815 in the testing group also indicated that our model functioned effectively in the prediction of tumor recurrence (Figure 5D and Figure 6D). As the patients’ risk scores increased, the number of deceased patients rose, which confirmed the accurate prognosis of the model for LUAD patients (Figure 5E and Figure 6E). The percentage of deceased patients in the high-risk group was about twice that in the corresponding low-risk group (Figure 5F and Figure 6F). For deceased patients, the survival time of the low-risk group was significantly longer than that of the high-risk group in both the training and testing groups. A similar phenomenon was also observed for living patients in the testing group (Figure 5G and Figure 6G). The results make sense because it is possible to be sure about the final survival time for deceased patients but not for living ones. In addition, all the assessments above confirmed that our risk score model could predict the prognosis of LUAD patients with high performance.

### 3.5. Knockdown of HAVCR1 Suppressed A549 Cell Proliferation

According to differential expression analysis, *HAVCR1* and *CYP24A1* were significantly up-regulated in LUAD, moreover, *HAVCR1* but not *CYP24A1* was identified as a prognostic gene for LUAD by our comprehensive bioinformatic analysis. Therefore, we designed siRNAs targeting *CYP24A1* and *HAVCR1* to investigate whether silencing the expression of *HAVCR1* and *CYP24A1* could affect cell proliferation in a LUAD cell line. The results of real-time PCR showed that *HAVCR1* and *CYP24A1* mRNAs were dramatically reduced by siRNAs (Figure 7A,B). CCK-8 assays were also conducted to assess the effects of *HAVCR1* and *CYP24A1* on the proliferation of A549 cells. The results demonstrated that the knockdown of *HAVCR1* suppressed nearly 20% of the proliferation of A549 cells while the knockdown of *CYP24A1* had no significant effect on the proliferation of A549 cells (Figure 7C,D), revealing that our prognostic model correctly selected the factors predicting LUAD.

## 4. Discussion

Lung adenocarcinoma is one of the most malignant cancers due to its poor prognosis. Increasingly, studies have made progress in unveiling the pathogenesis of LUAD and finding important biomarkers for the prognosis of LUAD. This gives a promising method to predict the survival time of patients and provides implications for further targeted therapeutic research.

Researchers have developed several risk score models to predict the prognosis of LUAD patients based on different biomolecules such as immune checkpoints [19], lncRNAs [20], ferroptosis-related genes [21], cuproptosis-related genes [22], DNA methylation [23], and RNA binding proteins [24]. Wang and colleagues constructed a prognostic model based on 13 cuproptosis-related ferroptosis genes (CRFGs), which were narrowed down from differentially expressed CRFGs in LUAD samples by univariate Cox regression analysis and LASSO Cox regression analysis, and included three genes associated with immune checkpoints. The AUC of the ROC curve was 0.716 in the training group and 0.629 and 0.742 in the testing groups [22]. Xu and colleagues focused on the genes with not only differential DNA methylation status on the promoter region but also differential mRNA expression in LUAD samples. Among them, 11 genes were selected through univariate and multivariate Cox regression analyses, and the features of promoter methylation from 11 genes were employed to construct a LUAD prognostic model. The Kaplan–Meier survival curve, ROC curve, and qPCR were conducted to support the model, and the AUC of the ROC curve was 0.697 [23]. Li et al. also performed univariate and multivariate Cox regression analyses and constructed a LUAD prognostic model composed of eight differentially-expressed RNA-binding proteins (RBPs). The AUC of the ROC curve was 0.775 and 0.814 in the training group and testing group, respectively [24]. However, the better accuracy in the testing group rather than in the training group reflected an underfitting problem in the model. 

In our study, we screened DEGs and prognosis-related mutant genes, then performed univariate and multivariate Cox regression analyses with these features, thereby identifying seven prognosis-related factors. The comprehensive validation confirmed that the seven prognosis-related factors were reliable in predicting the prognosis of LUAD patients. Compared to the previously-reported models, our model contained a variety of factors, including gene expression, gene mutant subtypes, and clinical information, rather than a single type of factor. Moreover, our model exhibited a relatively excellent performance to predict the prognosis for LUAD patients. 

In addition, we discovered that the knockdown of *HAVCR1* could delay the growth of A549 cells. *HAVCR1*, belonging to a mucin-like class I integral membrane glycoprotein, is highly expressed in tissues such as the kidney, colon, and rectum but has almost no expression in the lung, widely regulating the T-regulatory (Treg) cell activity through T-cell receptors [25,26]. *HAVCR1* has been reported to be involved in the allergic response, asthma, and the reduction in cell tight junction integrity [27]. Previous research has also revealed that *HAVCR1* was overexpressed in hepatocellular carcinoma and pancreatic adenocarcinoma and is possibly a marker for cancer [28]. To sum up, *HAVCR1* plays a crucial role in the immune system and probably in the progression of cancer. Our results are consistent with the previous report and support *HAVCR1* as an oncogene.

The protein domains and main biological functions of the prognostic genes in our model are displayed in Table 2. Some of them were also found to be related to tumorigenesis and LUAD progression. In A549 cells, miR-365b could suppress *GALNT4* expression to inhibit cell proliferation and colony formation and promote apoptosis by binding the 3’-UTR of *GALNT4* [17]. Previous research also identified *GRIA1* and *SATB2* as prognostic factors for LUAD [29,30]. In our results, the patients with a frameshift mutation or splice site mutation in *TP53* had the worst prognosis, while patients with a missense mutation or nonsense mutation had a more optimistic outcome. It is possible that the abnormal *TP53* proteins produced in a frameshift or splice site mutation allele might have a dominant negative effect, disrupting the normal *TP53* function from the other wild-type allele and having a more severe impact on the processes of DNA repair and apoptosis in cells. 

However, it is necessary to point out the limitations of this work. Initially, the construction and validation of the model were based on the TCGA database, lacking additional databases. In addition, the novel prognostic biomarkers need further experimental verification to elucidate their functional mechanisms in LUAD. Nevertheless, our study constructed a reliable prognostic risk score model for LUAD and discovered that the knockdown of *HAVCR1* inhibited A549 cell viability, forming the basis for further mechanistic and therapeutic studies in LUAD. 

## 5. Conclusions

In conclusion, this study focused on differentially expressed genes, different mutant subtypes of genes, and clinical information and constructed a seven-factor-based prognostic model. The model showed a remarkable performance in the prediction of overall survival, disease-free survival, and tumor recurrence. In addition, the newly-identified biomarker *HAVCR1* was proven to enhance the proliferation of LUAD cell line A549. Therefore, our risk score model has great potential to improve the evaluation of prognosis for LUAD patients and brings novel insight into targeted therapy.

## Figures and Tables

**Figure 1 diagnostics-13-01914-f001:**
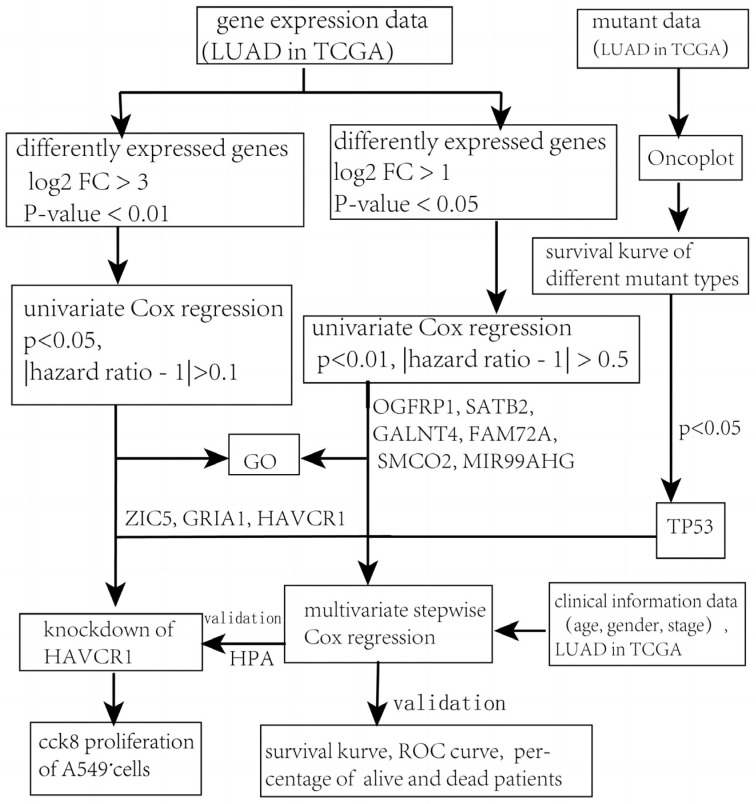
Flowchart of our study.

**Figure 2 diagnostics-13-01914-f002:**
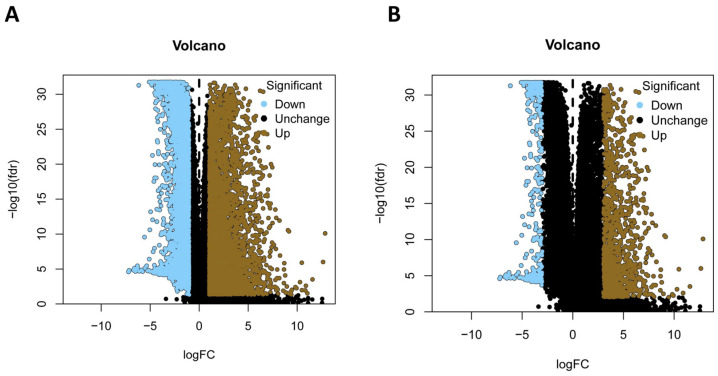
Identification of the differentially expressed genes (DEGs) between normal and LUAD samples. (**A**) Moderate threshold of DEGs of |log_2_ fold change (FC)| > 1 and false discovery rate(FDR) < 0.05. (**B**) Strict threshold of DEGs of |log_2_ FC| > 3 and FDR < 0.01.

**Figure 3 diagnostics-13-01914-f003:**
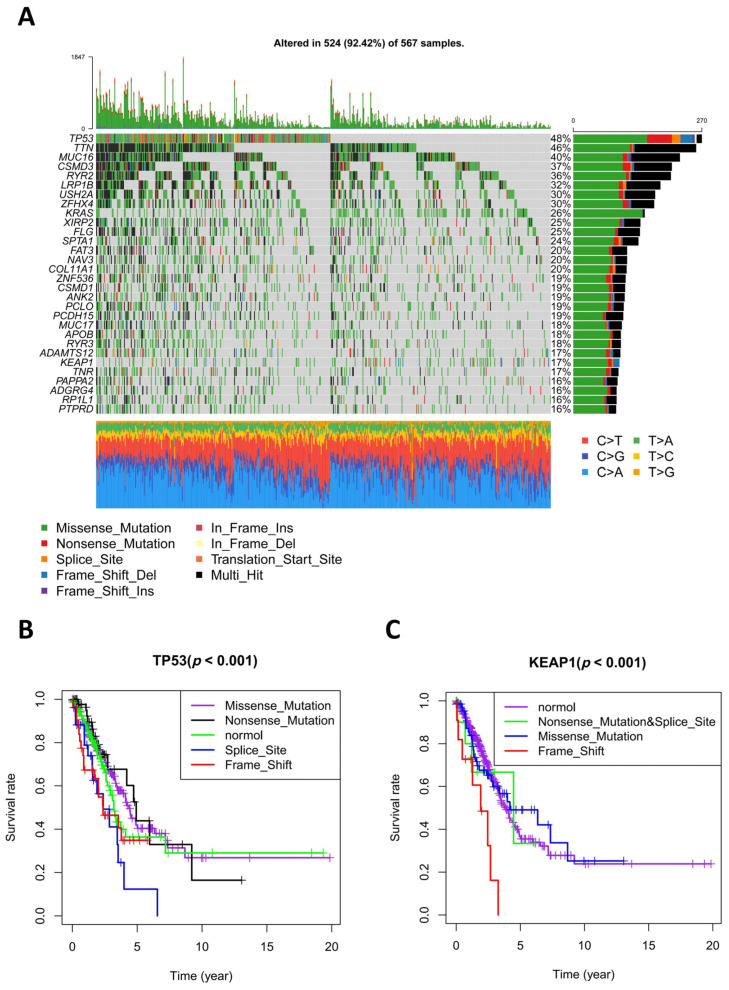
Identification of lung adenocarcinoma (LUAD) prognosis-related mutations from the Cancer Genome Atlas (TCGA) data. (**A**) Oncoplot displays the mutational patterns of the top 30 frequently mutated genes in 567 LUAD patients. In the Oncoplot chart, every column represents a patient, and the total mutational level of each patient is shown in the top histogram. Every row represents the mutational patterns of a gene from 567 patients: gray sites indicate no mutations and the other colors indicate different types of mutations. The total mutational level of each gene is displayed in the right histogram. The base substitution status of each patient is displayed in the bottom plot. (**B**) The comparison of survival rates among LUAD patients with different subtypes of *TP53*. (**C**) The comparison of survival rates among LUAD patients with different subtypes of *KEAP1*.

**Figure 4 diagnostics-13-01914-f004:**
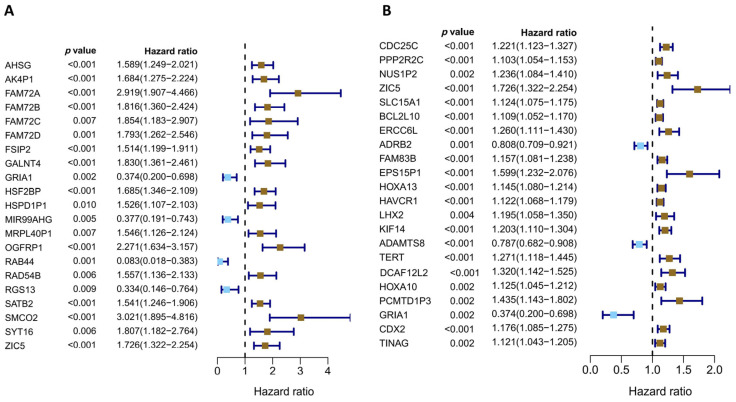
Forrest plotting of the univariate and multivariate Cox regression analyses in LUAD. Factors with a hazard ratio less than 1 are anti-risk factors for prognosis, and factors with a hazard ratio greater than 1 are risk factors for prognosis. (**A**) Forrest plotting of the univariate Cox regression analysis using DEGs identified with a moderate threshold. The genes with a *p*-value < 0.01, |hazard ratio − 1| > 0.5 in the univariate Cox regression analysis are shown. (**B**) Forrest plotting of the univariate Cox regression analysis using DEGs identified with a strict threshold. The genes with a *p*-value < 0.05, |hazard ratio − 1| > 0.1 in the univariate Cox regression analysis are shown. (**C**) GO enrichment analysis for the genes selected by univariate Cox regression. The GO terms contain the biological process (BP), cellular component (CC), and molecular function (MF). (**D**) Forrest plotting of multivariate stepwise Cox regression analysis with 13 candidate features using the data of the training group. (**E**) The formula of established prognostic risk score model.

**Figure 5 diagnostics-13-01914-f005:**
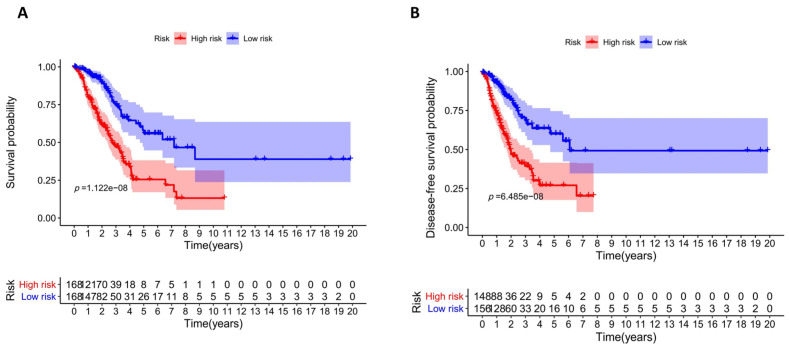
Validation of the training group indicates the good performance of our risk score prognostic model which includes seven factors. (**A**) Overall survival curve of the high-risk group and low-risk group; (**B**) disease-free survival curve of the high-risk group and low-risk group; (**C**) ROC curve of the survival state; (**D**) ROC curve of tumor recurrence; (**E**) survival status and risk scores; (**F**) percentages of living and deceased patients in the low-risk group and high-risk group; and (**G**) the average survival time of living and deceased patients in the high-risk and low-risk groups.

**Figure 6 diagnostics-13-01914-f006:**
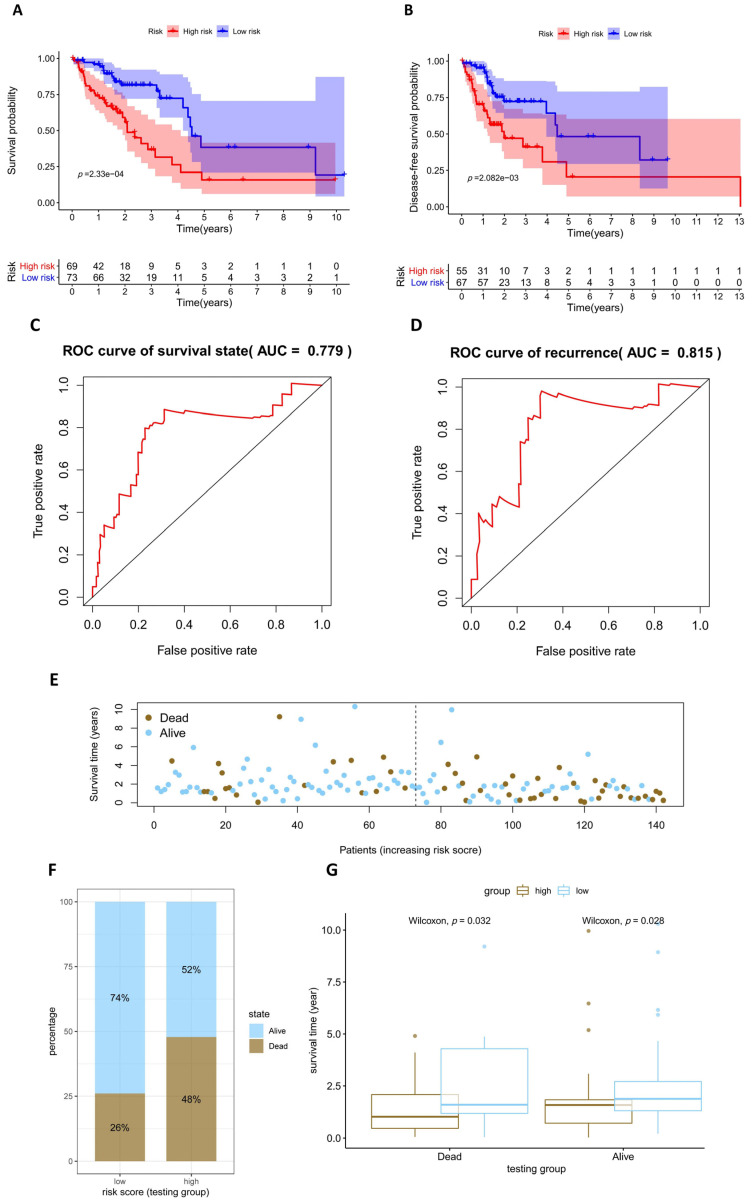
Validation of LUAD patients from the testing group indicates the good performance of our risk score prognostic model established with seven factors. (**A**) Overall survival curve of the high-risk group and low-risk group; (**B**) disease-free survival curve of the high-risk group and low-risk group; (**C**) ROC curve of the survival state; (**D**) ROC curve of tumor recurrence; (**E**) survival status and risk scores; (**F**) percentages of living and deceased patients in the low-risk group and high-risk group; and (**G**) the average survival time of living and deceased patients in the high-risk and low-risk groups.

**Figure 7 diagnostics-13-01914-f007:**
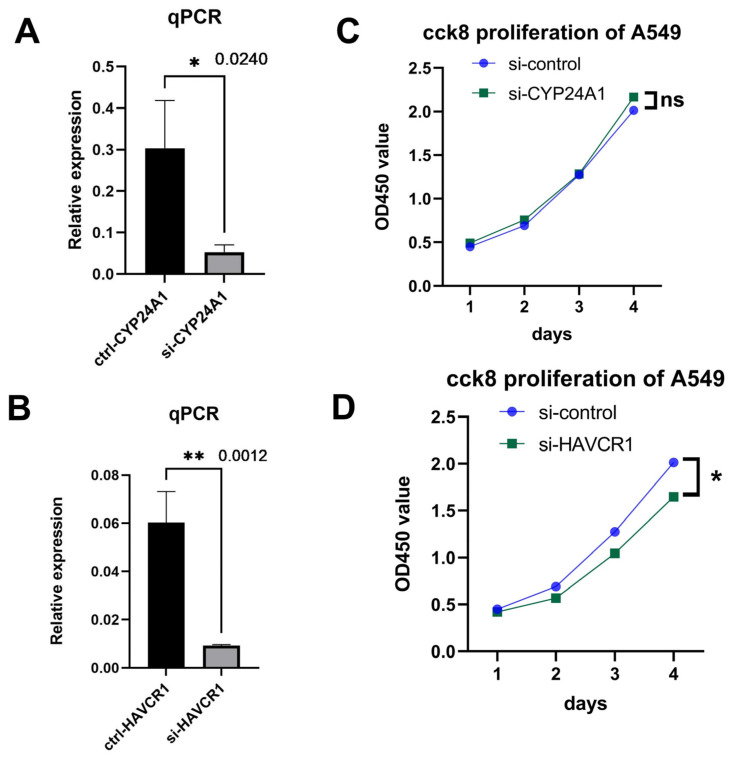
The effect of *CYP24A1* or *HAVCR1* knockdown on A549 cell proliferation. (**A**,**B**) The results of quantitative PCR showing the siRNAs significantly depleted the mRNAs of *CYP24A1* and *HAVCR1*. (**C**,**D**) CCK-8 assay after the knockdown of *CYP24A1* or *HAVCR1* in A549 cells indicating that the *CYP24A1* knockdown does not affect cell growth while the knockdown of *HAVCR1* inhibits cell proliferation. *n* = 3, *p*-values examined by one-tail *t*-test, * *p* < 0.05, ** *p* < 0.01.

**Table 1 diagnostics-13-01914-t001:** The coefficients of the seven factors obtained by the multivariate stepwise Cox regression.

ID	Coefficient	HR (Hazard Ratio)	HR.95L	HR.95H	*p*-Value
*SMCO2*	0.737888	2.091513	1.127194	3.88081	0.019305
*SATB2*	0.357248	1.429391	1.081459	1.88926	0.012065
*HAVCR1*	0.061489	1.063419	0.998669	1.132368	0.05506
*GRIA1*	−0.72351	0.485047	0.220938	1.064871	0.07134
*GALNT4*	0.43393	1.543311	1.049862	2.268687	0.027279
*TP53*	0.14491	1.155935	0.954653	1.399657	0.137666
stage	0.353671	1.424287	1.193488	1.699717	8.82 × 10^−5^

**Table 2 diagnostics-13-01914-t002:** The protein domains and biological functions of the six genes in the model.

Gene	Protein Domains in Pfam	Biological Functions
*SMCO2*	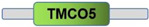	The function is unknown. A single-pass membrane protein with two coiled-coil domains (NCBI: https://www.ncbi.nlm.nih.gov/gene/341346/ortholog/?scope=32525, accessed on 27 March 2023).
*SATB2*	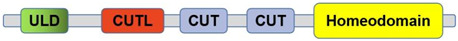	Controls the transcription factor expression of nuclear genes by binding to the matrix attachment regions of DNA. A docking site for several chromatin remodeling enzymes. Recruits corepressors (HDACs) or coactivators (HATs) to promoters and enhancers. (HPA: https://www.proteinatlas.org/ENSG00000119042-SATB2, accessed on 27 March 2023)
*HAVCR1*	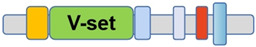	Cell recognition, immune activation, tight junction, and cancer biology. Promotes the activation and proliferation of immune cells and cytokine secretion and regulates the function of NK cells and CD^8+^ T for an efficient antitumor immune response [28].
*GRIA1*	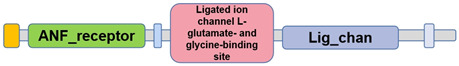	An ionotropic glutamate receptor that has been proven to be related to neurotransmitter systems, sleep, and circadian rhythm [31].
*GALNT4*	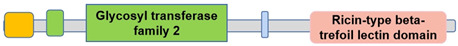	Initiates the mucin-type O-glycosylation in cells. Participates in the post-translational modification, invasion, and proliferation of cancer cells [17,32].
*TP53*	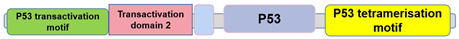	A tumor suppressor protein with oligomeric domains that participates in the regulation of target gene expression, cell cycle arrest, apoptosis, senescence, changes in metabolism, and DNA repair. Mutations of this gene are related to various cancers (NCBI: https://www.ncbi.nlm.nih.gov/gene?Cmd=DetailsSearch&Term=7157, accessed on 27 March 2023).

## Data Availability

The data we used in bioinformatic analyses were obtained from the Cancer Genome Atlas (TCGA, https://portal.gdc.cancer.gov/, accessed on 29 December 2020) public database.

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
