# Peer review of "Identification of New Prognostic Genes and Construction of a Prognostic Model for Lung Adenocarcinoma"

_diagnostics, 2023, doi:10.3390/diagnostics13111914_

Round 1

Reviewer 1 Report

1. The drawings are very small, the inscriptions on the drawings are unreadable. All drawings are in need of editing.

2. The proposed predictive model takes into account only gene expression. Why did the authors not combine gene expression data with other information about the patient: adenocarcinoma stage, lymph node involvement, presence of mutations, comorbidities, etc.? In this case, the predictive value would increase significantly?

3. It would be interesting to compare the prognostic data of the authors with other prognostic models, if they exist.

4. The proposed model describes overall survival, but not disease-free survival. Is it possible to estimate the likelihood of recurrence based on gene expression data?

Reviewer 2 Report

1. Two first sentences in abstract should be reject.

2. Genes should be written in italics.

3. In introduction first paragraph including references form 1 to 6 should removed or modified. 

4.There is no information about material but in section 2.6 there is part about cell line. Also in section 2.6 there are mixed a lot of elements. It should be divided and better described.

5. The discussion should be also modified, because there is lack of disscusion of obtained results to the state of art in this field.

Round 2

Reviewer 1 Report

The authors have responded to the comments of the reviewers, I have no more questions. However, the quality of the drawings, especially the inscriptions, remains unsatisfactory.

Reviewer 2 Report

The manuscript was corrected according to sugesstions.